# Rapid learning of a phonemic discrimination in the first hours of life

Yan Jing Wu [1,8], Xinlin Hou[2,8], Cheng Peng [2], Wenwen Yu [3], Gary M. Oppenheim [4], Guillaume Thierry [4,5] and Dandan Zhang [3,6,7] ✉

**Human neonates can discriminate phonemes, but the neural mechanism underlying this ability is poorly understood. Here we show that the neonatal brain can learn to discriminate natural vowels from backward vowels, a contrast unlikely to have been learnt in the womb. Using functional near-infrared spectroscopy, we examined the neuroplastic changes caused by 5 h of post-natal exposure to random sequences of natural and reversed (backward) vowels (T1), and again 2 h later (T2). Neonates in the experimental group were trained with the same stimuli as those used at T1 and T2. Compared with controls, infants in the experimental group showed shorter haemodynamic response latencies for forward vs backward vowels at T1, maximally over the inferior frontal region. At T2, neural activity differentially increased, maximally over superior temporal regions and the left inferior parietal region. Neonates thus exhibit ultra-fast tuning to natural phonemes in the first hours after birth.**

Human neonates have remarkable linguistic sensitivity and the ability to process elaborate speech stimuli within hours of being born. At birth, they show preferences to speech sounds over a variety of non-linguistic, complex sounds[1,2]. They also prefer their mother's voice compared with other female voices[3]. An important aspect for our understanding of language ability in neonates is phoneme discrimination, such as the ability to discriminate vowels[4] and syllables[5] (for example, consonant–vowel combinations). Given that phonemes are the smallest discriminable units of speech sounds[6], phoneme discrimination reveals the fundamental perceptual sensitivity that supports the development of speech perception in the future. It is different from the more general auditory disposition that allows neonates to process acoustic rather than linguistic aspects of speech, such as rhythm[7], sound duration[8], temporal relations between syllables[9,10] (for example, sequence and repetition), stress patterns in multisyllabic words[11] and acoustic characteristics of utterances[12].

It is generally believed that neonates can discriminate phonemes in most languages at birth before 'tuning into' the specific phonemic categories used in their native language over the first few months[13–15]. For instance, the sucking rate of Swedish and American neonates was measured after they had been presented with native and non-native vowels[16]. Sucking amplitude increased when the infants heard vowels of an unfamiliar non-native language, as compared with when they heard vowels from their native language, suggesting not only sensitivity to phonemes in both languages but also experiential influences on vowel perception, indicating an effect of prenatal learning. This finding is consistent with evidence for the auditory system becoming operational as early as 24 weeks into gestation[17], allowing exposure to spoken language in utero to start shaping the characteristics of auditory perceptual representations[1] and, in particular, speech representations[18]. Interestingly, the effect of prenatal learning on vowel perception appears to be independent from neonates' postnatal contact with the ambient (that is, the native) language, since no difference was found between

participants tested at 7 and 75 h after birth[16]. Although 75 h is a relatively short period of time for postnatal learning, these findings suggest that the neonatal speech perception system has already attained a certain level of maturation and crystallization at birth.

In contrast, the seminal study by Cheour et al.[19] showed that exposure to speech sounds can affect the neural dynamics associated with phoneme discrimination immediately after birth. The authors presented natural vowels that exist in most human languages to neonates during a 2.5–5 h training session, inserted between a baseline test and a post-training test during which they measured the amplitude of the mismatch negativity[20] (MMN). MMN is a neurophysiological index of automatic change detection in the auditory input used as an effective tool to investigate the neural dynamics of passive learning in infants[21] and newborns (see ref. [22] for a magnetoencephalography equivalent). Cheour et al.[19] showed that, while there was a significant MMN response to the acoustically simple vowel sound /i/ (deviant) presented among /y/ (standards) before and after training, the MMN response to the acoustically complex vowel sound /y/i/ reached statistical significance only after training. This pattern of results persisted for at least 24 h post training and generalized to the same vowels presented at a different pitch, suggesting that short-term (<5 h) exposure to speech sounds can affect phonemic perception ex utero.

While vowel discrimination in early infancy has been demonstrated previously, little is known regarding the neural mechanisms and dynamics associated with postnatal phonological learning immediately after birth. Here we used functional near-infrared spectroscopy (fNIRS) to first assess neonatal phoneme perception within 3 h of birth and then measure neuroplastic changes induced by postnatal exposure to natural (forward) and reversed (backward) vowels over the following 7 h. NIRS has a relatively high spatial resolution compared with other non-invasive methods compatible with neonate testing, such as electroencephalography (EEG). Its high motion tolerance makes it ideally suited to test very young infants[9,10,23–25]. We used strings of naturally produced vowels in the

[1]Faculty of Foreign Languages, Ningbo University, Ningbo, China. [2]Department of Pediatrics, Peking University First Hospital, Beijing, China. [3]School of Psychology, Shenzhen University, Shenzhen, China. [4]School of Psychology, Bangor University, Bangor, Wales, UK. [5]Faculty of English, Adam Mickiewicz University, Poznań, Poland. [6]Institute of Brain and Psychological Sciences, Sichuan Normal University, Chengdu, China. [7]Shenzhen-Hong Kong Institute of Brain Science, Shenzhen, China. [8]These authors contributed equally: Yan Jing Wu, Xinlin Hou. ✉e-mail: zhangdd05@gmail.com

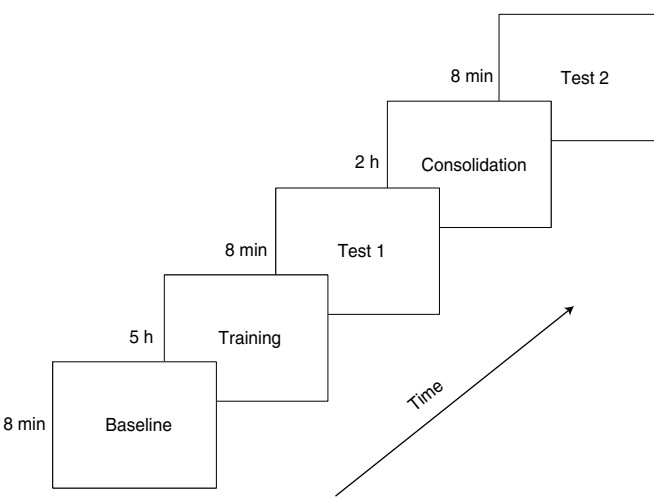

**Fig. 1 | Schematic of the experimental procedure.** fNIRS data were recorded at onset (T0, baseline), 5 h later (T1) and another 2 h later (T2). Training involved exposure to forward and backward stimuli in blocks, and test sessions involved random presentations of a specific set of vowels pronounced naturally or played backwards. In the consolidation phase, neonate participants were at rest and received no stimulation.

native language (Mandarin Chinese; for example, /ɑː/, /ɔː/, /iː/, /uː/, /əː/, and /æ/) and the same auditory sequences played backwards to serve as acoustically matched control stimuli to minimize prosodic variation confounds[26] and the likelihood of such contrast having been learnt before birth. Moreover, we used steady-state vowels rather than syllables (for example, consonant–vowel combinations) to avoid possible distortion by consonantal context of vowel discrimination, relating to voicing and place of articulation[5].

First, we collected baseline fNIRS data in response to speech and non-speech stimuli presented randomly for 8 min (T0). Then, two groups of neonates (experimental participants and active controls) were exposed to 10-min blocks of forward and backward vowels presented in alternation for 5 h (/ɑː/, /ɔː/ and /iː/ for the experimental group; /uː/, /əː/ and /æ/ for the active control group). Another group of control participants (passive controls) received no specific stimulation or training during the following 5 h but were placed in the same environment as the experimental and active control participants. Immediately after the end of the training session (or equivalent resting period for the passive controls), a test (T1) was carried out in all three groups and 2 h later, a final measurement was taken (T2) to test for potential consolidation effects (Fig. 1). The stimuli, fNIRS acquisition procedure and measurements used at T1 and T2 were identical to those used at T0 but, while the same vowels were used for training and testing in the experimental group, active control participants were trained with different vowels (and corresponding reverse vowels; Supplementary Table 2). The entire experiment was completed within the first 24 h of the participants' birth.

Despite the subtlety of the contrast tested, we anticipated that all three groups of participants would be able to dissociate forward from backward vowels at T0 due to prenatal exposure to natural vowels in the womb[16]. Further, we expected language-specific perceptual learning to occur only in the experimental group, with training of the contrast between forward and backward vowels eliciting enhanced phonological contrast sensitivity at T1 and T2 relative to both active and passive controls. This hypothesis was based on the premise that vowel perception tuning is highly specific and thus does not generalize across different vowels[19]. Beyond testing phonemic tuning specificity, we aimed to characterize the neuroanatomical substrates involved in the early perception of a subtle

phonological contrast. We specifically anticipated involvement of the superior temporal (ST) and inferior frontal (IF) brain regions, since they have previously been associated with processing spoken language in neonates and babies[24,27–31]. We also examined changes in resting-state functional connectivity between testing sessions to explore interactions between key regions in the neural network involved. We expected an increase in resting-state functional connectivity between T0 and T1 in both experimental and active control participants relative to passive controls, given that both these groups had been exposed to a vowel contrast in the interval between T0 and T1.

## Results

**Oxyhaemoglobin concentration amplitude analysis.** A linear mixed effects regression analysis of the per-trial mean oxyhaemoglobin concentration [HbO] amplitudes from 6–16 s post stimulus onset (summarized in Supplementary Table 1; see Methods for full details) identified a significant super-additive three-way interaction between stimulus type, the second participant group contrast (active control vs experimental) and the second phase contrast (T1 vs T2), indicating that the difference in [HbO] between forward and backward vowel conditions was greater for the experimental group compared with the active control group, and in the post-training, post-consolidation assessment compared with the post-training, pre-consolidation assessment ($\beta = 0.125\,\mu$mol l$^{-1}$, s.e.m. $= 0.058$, $t(86.7) = 2.15$, $\sim P = 0.034$).

All other significant effects in the analysis involved subcomponents of this three-way interaction and appeared to be driven by it. Plotting the best linear unbiased predictors (BLUPs) for the three-way interaction suggested a bilaterally symmetric distribution that was maximal over the superior temporal and supramarginal (SM) regions bilaterally, and in the inferior parietal (IP) region that was somewhat stronger in the left hemisphere (Fig. 2 and Table 1).

**Oxyhemoglobin concentration peak analysis.** Next we used the same approach to consider temporal aspects of [HbO] variation over time. Although researchers seldom consider the timecourse of fNIRS measurements, such analyses have long been reported to convey meaningful information in the case of other oxygenation-based measures with slow timecourses (for example, time-resolved fMRI, see for instance ref. [32]). Our 10 Hz measurement rate provides a suitable basis for analysing the peak latency of [HbO] over time (Methods). We first identified the latency of the [HbO]$_{max}$ in each trial and then fitted the same linear mixed effects regression model to the latency data that we had previously fitted to the mean amplitude data. This regression identified a significant three-way interaction between stimulus type, the second participant group contrast (active control vs experimental) and the first test phase contrast (T0 vs mean (T1,T2); Supplementary Table 1), indicating that the difference in [HbO] peak latencies associated with forward versus backward vowel stimuli was greater for the experimental group compared with the active control group, and in the post-training assessment than in the pre-training assessment ($\beta = -0.569\,\mu$mol l$^{-1}$, s.e.m. $= 0.209$, $t(95.0) = -2.72$, $\sim P = 0.008$). As was the case for the amplitude analysis above, all other significant effects in this latency analysis appeared to be driven by this three-way interaction.

Plotting the BLUPs for the three-way interaction suggested an anterior distribution that was maximal over the inferior frontal regions bilaterally (Fig. 3 and Table 2).

**Functional connectivity analysis.** We then analysed resting-state functional connectivity between brain regions. To focus on the most relevant channels, we first selected as seeds 7 channels (that is, 2, 6, 7, 10, 43, 44 and 45) from the amplitude and latency analyses that survived false discovery rate (FDR) correction ($q < 0.15$)[33,34], and then correlated the 10 Hz measurements between these and all other

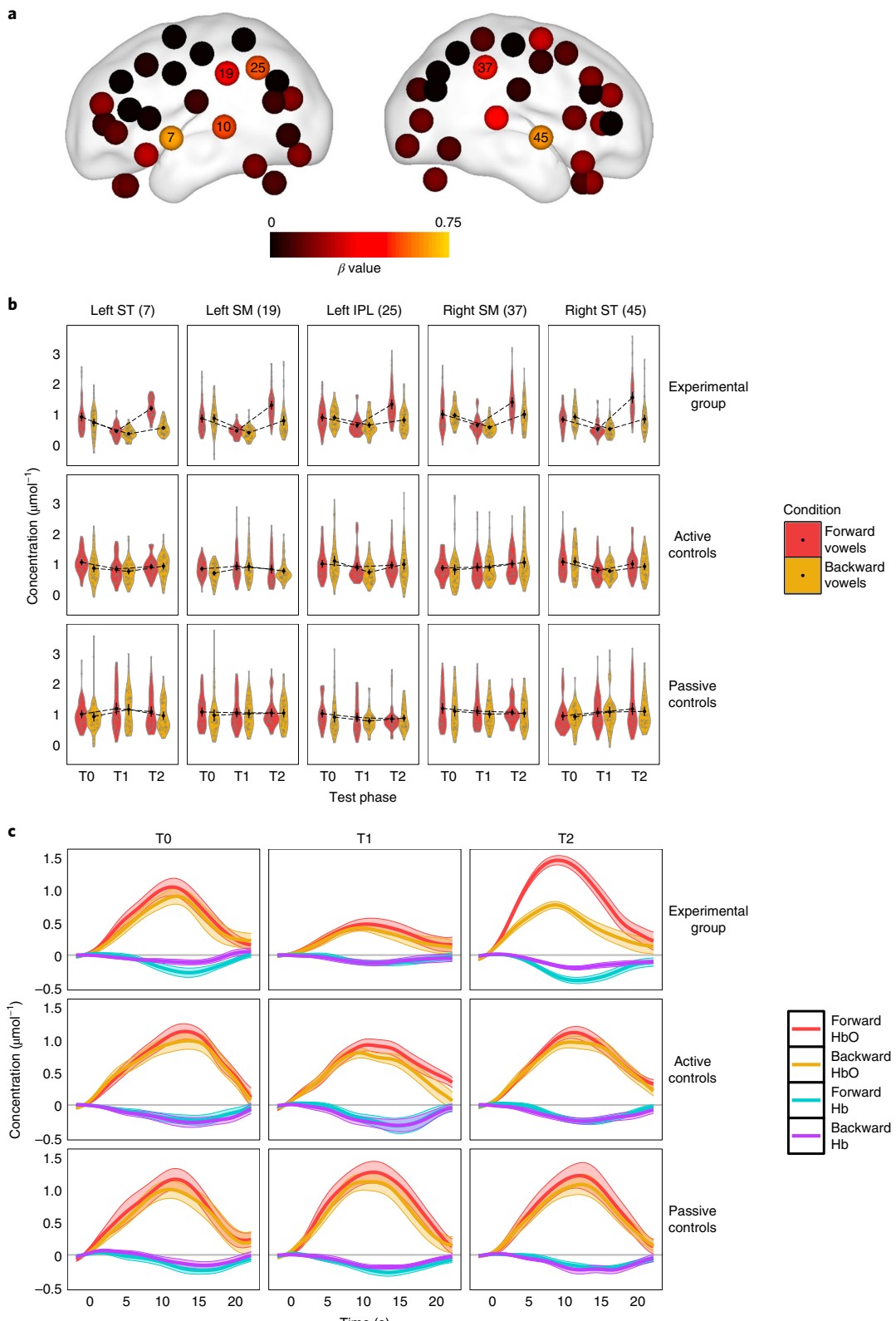

**Fig. 2 | [HbO] mean amplitude results. a**, Plot of $\beta$ estimates for the BLUPs of the three-way interaction between group contrast (active control vs experimental), stimulus type (forward vs backward) and test phase contrast (T1 vs T2) on [HbO] mean amplitude. $\beta$ values are plotted per channel on a neonate brain model (37 weeks) elaborated by ref. [94], using the BrainNet Viewer toolbox[95]. **b**, Violin plots of observed [HbO] values in response to forward and backward vowels in 5 of the channels listed in Table 1 (results for channel 10 (not pictured) closely resembled those illustrated for channel 7). Experimental group $n = 22$, active control group $n = 23$ and passive control group $n = 21$. Black dots depict means and error bars display 95% confidence intervals. Brain regions are labelled according to the abbreviations used in the main text. **c**, Representative examples of [HbO] and [Hb] variation over time in each of the three groups and test sessions in channel 7 set over the left ST region. Waves depict mean concentration evolution over time averaged across individual data, bounded by s.e.m. in the corresponding transparent shade.

**Table 1 | Individual channels where the crucial three-way interaction (group (contrast 2) × stimulus type × phase (contrast 2)) for mean [HbO] amplitude reached significance**

| Channel | Region | β | s.e.m. | d.f. | t | P |
| --- | --- | --- | --- | --- | --- | --- |
| 7 | Left superior temporal | 0.679 | 0.220 | 63.00 | 3.09 | 0.003 |
| 10 | Left superior temporal | 0.519 | 0.190 | 63.00 | 2.73 | 0.008 |
| 25 | Left inferior parietal | 0.554 | 0.214 | 63.00 | 2.59 | 0.012 |
| 19 | Left supramarginal | 0.409 | 0.187 | 63.00 | 2.19 | 0.032 |
| 37 | Right supramarginal | 0.452 | 0.217 | 62.99 | 2.09 | 0.041 |
| 45 | Right superior temporal | 0.653 | 0.227 | 63.00 | 2.88 | 0.005 |

The BLUPs for the three-way interaction of the linear mixed effects regression analysis. β, beta estimate; s.e.m., standard error of the mean; d.f., degrees of freedom; t, t-value (two-tailed test), P = estimated P value (uncorrected for multiple comparisons).

channels for each subject in the 3 min (1,800 samples) before each test[35], using a Fischer z-transformation (that is, $z = \text{artanh}(r)$) to bring the Pearson correlation coefficients into a normal distribution. (Note: the results of FDR correction were largely insensitive to the choice of threshold. With the threshold reduced to $q < 0.05$, which restricted the seed channels to 2, 6, 43 and 44, the crucial interaction remained essentially the same: $= 0.221$, $P = 0.002$. If including all channels ($\sim q \leq 1$), it was estimated as $= 0.161$, $P = 0.001$). Figure 4 illustrates the mean correlation coefficients for each group at each assessment. To assess changes in the strength of these connections as a function of experience, we applied reduced forms of the same linear mixed effects models that were used for the amplitude and latency analyses. This regression, summarized in Table 3, identified a significant two-way interaction between the first participant group contrast (passive control vs mean (active control, experimental)) and the second phase contrast (T1 vs T2), indicating stronger increases in connectivity for the two groups that received auditory training, specifically after sleep ($\beta = 0.217$, s.e.m. $= 0.062$, $t(50.2) = 3.48$, $\sim P = 0.001$). All other significant effects in this analysis appeared to be driven by this two-way interaction.

This interaction would be considered significant at an uncorrected $\alpha = 0.05$ for 32 out of 336 pairs of channels (10%; 30 positive, 2 negative; Fig. 4), often involving channels in the vicinity of the left IF (chan. 2: 9 pairs, chan. 6: 6 pairs), left ST (chan. 7: 11 pairs, chan. 10: 4 pairs), right IF (chan. 43: 2 pairs, chan. 44: 3 pairs) and right ST regions (chan. 45: 4 pairs). Notably, these included multiple connections between channels over the left IF and left ST regions (chan. 6 and 7: $\beta = 0.886$, s.e.m. $= 0.398$, $t(49.0) = 2.22$, $\sim P = 0.031$; chan. 2 and 5: $\beta = 1.078$, s.e.m. $= 0.348$, $t(49.0) = 3.10$, $\sim P = 0.003$; chan. 2 and 7: $\beta = 1.078$, s.e.m. $= 0.380$, $t(49.0) = 2.84$, $\sim P = 0.007$; chan. 4 and 7: $\beta = 0.856$, s.e.m. $= 0.403$, $t(49.0) = 2.13$, $\sim P = 0.038$; chan. 2 and 10: $\beta = 0.792$, s.e.m. $= 0.388$, $t(49.0) = 2.04$, $\sim P = 0.047$), left IF and left IP regions (chan. 6 and 25: $\beta = 1.078$, s.e.m. $= 0.354$, $t(49.0) = 3.04$, $\sim P = 0.004$; chan. 2 and 25: $\beta = 1.131$, s.e.m. $= 0.383$, $t(49.0) = 2.95$, $\sim P = 0.005$), and left ST and right ST regions (chan. 7 and 45: $\beta = 0.905$, s.e.m. $= 0.432$, $t(49.0) = 2.10$, $\sim P = 0.041$).

## Discussion

Neonates can already discriminate some phonemes at birth, apparently on the basis of in utero auditory learning. However, little is known about the plasticity of the neonatal phonological system immediately after birth, and it is unclear whether newborns can already distinguish subtle variations between vowels[19]. Here, using fNIRS sensors distributed around neonates' scalp, we examined amplitude and latency variations in haemoglobin concentration elicited by forward vowels and their waveform reversal. Linear mixed effects regressions revealed overall three-way interactions for both fNIRS mean amplitude and peak latency, indicating that experimental participants, as compared with active controls, discriminated between forward and backward vowels faster and to a

greater extent after prolonged exposure to such stimuli. Given that both experimental and active control participants were exposed to (different sets of) vowels in the training session, and that there was no indication of such changes in active control participants, the interactions most probably reflect neural mechanisms associated with specific vowel acquisitions, rather than general experience with speech sounds. The learning effect on mean amplitude specifically emerged when comparing pre- and post-consolidation sessions, where the experimental group showed increased mean [HbO] amplitudes in response to forward as compared with backward vowels at T2 (Fig. 2), producing the largest estimates for sensors placed above the superior temporal (channels 7 and 45) and supramarginal regions (channels 19 and 37) bilaterally, and the left inferior parietal region (channel 25). Research in adults has associated the ST gyrus with cognitive processes underlying speech comprehension and in particular the extraction of speech sounds from complex auditory input. For instance, in a study of the 'cocktail party effect', bilateral ST activation was observed when an attended speech stream was successfully extracted from background noise, whereas disruption of the attended speech stream by background noise resulted in left-lateralized activation[36,37]. Studies of neonates and infants have also implicated the ST regions in early auditory language comprehension, for instance in relation to phonological processing[38] and the processing of affective prosody[26].

The left IP region has also been implicated in adult speech comprehension[39,40]. Left lateralization is a sign of in-depth neural specialization during language development[41] and a consensus has yet to be reached regarding the ubiquity of left lateralization for language processing in neonates. It is worth noting, however, that the head montage used in the present study attempted to segregate activities recorded over IP, SM and angular regions (Supplementary Table 3), in keeping with mainstream protocols for neuroimaging research in neonates[42–44]. The learning effect was maximal on channels located above the SM and angular regions, which play a critical role in phonological and semantic processing of words, respectively[45]. For example, the SM gyrus is often considered part of Wernicke's area in adults, and is known to support phonological processing during listening comprehension through analysis of the temporal information embedded in words and speech segmentation[46,47]. Lesions in the SM gyrus can cause receptive aphasia characterized by a loss of the ability to relate spoken and written words to one another, and the SM region is thought to enable covert articulation of words (that is, subvocal repetition[48,49]). Therefore, the combined observation of activations in fNIRS channels located approximately above the left IP and SM regions suggests that hearing vowels may prompt a nascent, neonatal imitation network into action, that is, the activation of the perception-production loop known to become critical for language learning in later life.

The learning effect for peak latency was observed when comparing pre- and post-training sessions: the experimental group showed

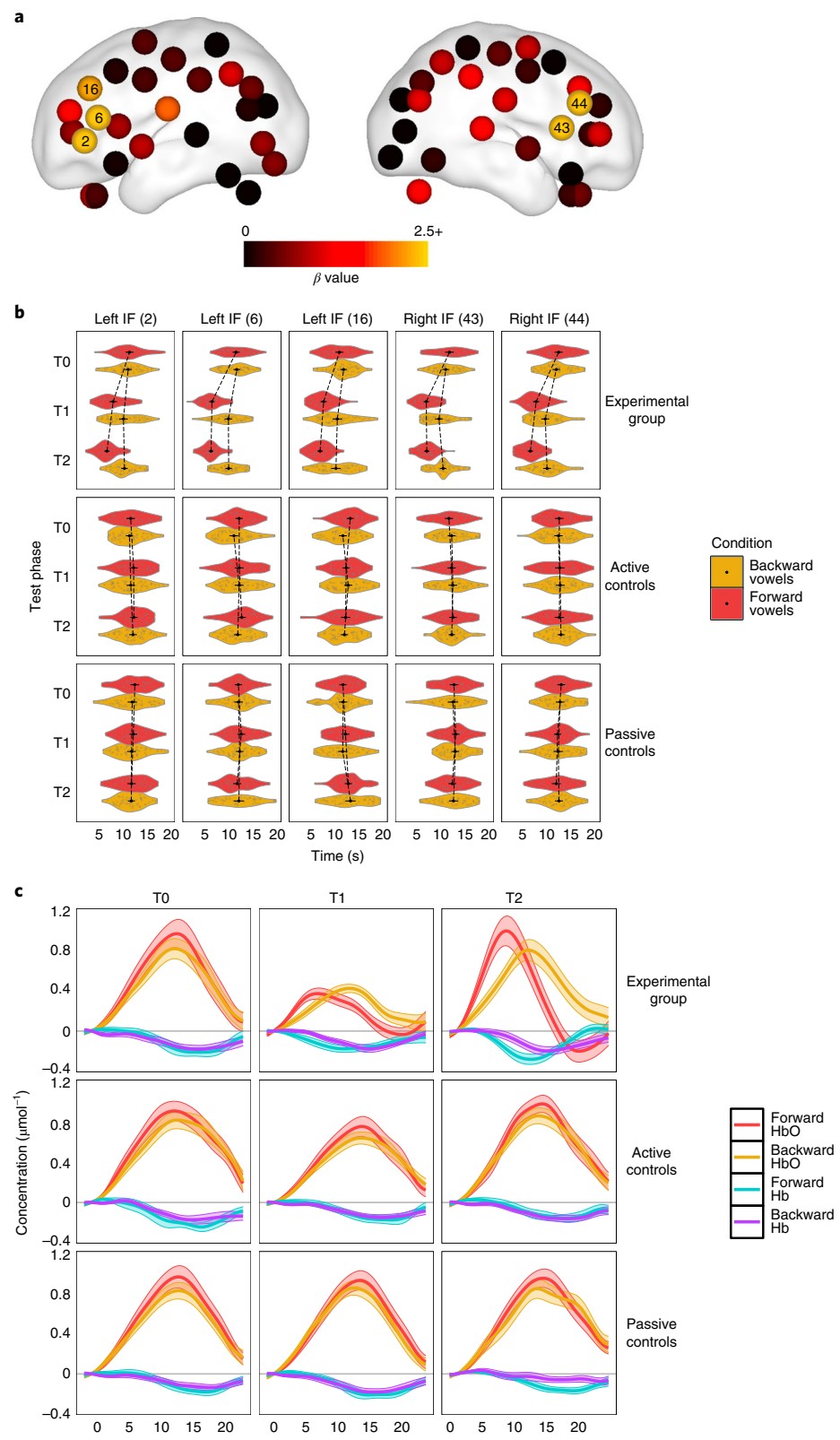

**Fig. 3 | [HbO] peak latency analysis results. a**, Plot of $\beta$ estimates for the BLUPs of the three-way interaction between the active control vs experimental group contrast, the stimulus type contrast (forward vs backward) and the T0 vs mean (T1, T1) phase contrast on [HbO] mean peak latency. $\beta$ values are plotted per channel on a neonate brain model (37 weeks) elaborated in ref. [94], using the BrainNet toolbox[95]. **b**, Violin plots of observed [HbO] peak latencies in response to forward and backward vowels for channels listed in Table [2]. Experimental group $n = 22$, active control group $n = 23$ and passive control group $n = 21$. Black dots depict means and error bars display 95% confidence intervals. **c**, Representative examples of [HbO] and [Hb] variation over time in each of the three groups and test sessions in channel 6 set over the left IF region. Waves depict mean concentration evolution over time averaged across individual data, bounded by s.e.m. in the corresponding transparent shade.

**Table 2 | Individual channels where the crucial three-way interaction (group (contrast 2) × stimulus type × phase (contrast 1)) for peak [HbO] latency reached significance**

| Channel | Region | β | s.e.m. | d.f. | t | P |
|---|---|---|---|---|---|---|
| 2 | Left inferior frontal region | −2.60 | 0.84 | 63.01 | −3.09 | 0.003 |
| 6 | Left inferior frontal region | −2.96 | 0.63 | 63.00 | −4.70 | <0.001 |
| 16 | Left inferior frontal region | −2.15 | 0.87 | 63.00 | −2.48 | 0.016 |
| 43 | Right inferior frontal region | −3.79 | 0.82 | 63.00 | −4.65 | <0.001 |
| 44 | Right inferior frontal region | −2.87 | 0.69 | 63.00 | −4.14 | <0.001 |

The BLUPs for the three-way interaction of the linear mixed effects regression analysis are as in Table 1.

reduced peak latencies in response to forward as compared with backward vowels at T1 and T2 (Fig. 3), with an anterior distribution over the inferior frontal regions bilaterally (for example, channels 6 and 44). In adults, the IF gyrus is considered part of the auditory dorsal stream and has been associated with language processing, especially speech production (that is, Broca's area[50–52]), speech comprehension[53,54] and speech monitoring[55]. Studies conducted in neonates and infants have associated activations in the IF region with discrimination of speech sounds[56–59] and a predictor of future language ability[60]. Therefore, our findings suggest that distinguishing between forward and backward vowels entails a change in neural effectiveness, resulting in latency changes, reminiscent of changes in the timecourse of the blood-oxygen-level dependent (BOLD) signal in fMRI observed in adults[32,61,62].

However, the forward–backward vowel contrast seemed to have elicited only minimal variations in all groups of participants at T0 (Figs. 2 and 3), suggesting that neonates might not have been able to distinguish between the two categories of stimuli before exposure. It is worth noting that the present study used a single token for each vowel sound, played forward and backward, which is a different approach to that taken in previous studies[16]. As indicated by prosodic rating results (Methods), our stimuli entailed minimal prosodic variations, which might in part account for the discrepancy between our results and previous findings of neonatal discrimination in the case of more complex speech stimuli (for example, story-telling[24,63]). While speech signals heard in the womb are filtered, neonates differentially respond to some prosodic contrasts at birth, implying that some prosodic information reaches the foetus[26,64]. However, it is highly improbable that the subtle distinction between forward and backward vowels used in the present study would have survived filtering through the mother's tissues and placenta, which helps explain why our participants might have failed to distinguish between the two stimuli immediately after birth. It is all the more remarkable then that merely after 5 h of exposure, we could see specific discrimination emerge.

In comparison with the learning effect on peak latency, which was observed immediately after exposure, the effect on mean amplitude was observed after 2 h of rest following the 5 h of exposure to the critical vowel. This difference might suggest that it is easier to fire neurons earlier than producing a sustained increase in the magnitude of neural activation, which requires a more effortful process. We speculate that neuroplastic changes yielded by the training session were consolidated during the 2 h resting period separating T1 and T2, during which neonates were mostly asleep, as shown by polysomnography data (Supplementary Table 4). Such an account is consistent with research investigating links between sleep and memory formation[65]. Following acquisition of a new piece of information or skill, sleep allows consolidation without the need for further exposure or training. Sleep-mediated memory (synaptic) consolidation has been shown to be critical for the formation of phonetic representations[66] and language acquisition more generally[67]. While previous studies have mostly focused on adults and infants (but see ref. [19]), our findings show that consolidation through sleep readily applies to perceptual learning as soon as we are born.

By examining the effects of short-term training on neurophysiological correlates of speech perception, we were able to show that human neonates are predisposed to acquire speech: while neuroplastic changes take place following a short period of exposure to auditory stimuli that are fundamental for human speech recognition (that is, vowels), no such effects were obtained for unnatural, backward vowels. This finding is consistent with evidence showing that neonates start learning to dissociate words differing by a single vowel as soon as they are born[68]. Most importantly, the three-way interaction found in the present study was maximal at fNIRS channels placed above IF, ST and SM regions bilaterally, allowing the characterization of a speech acquisition network reminiscent of the mirror neuron network described in adults (encompassing Broca's area, the ST gyrus and the left IP lobule). These regions have been implicated in the identification and imitation of speech-related actions in others, a fundamental process in language acquisition that connects speech perception with production[69,70]. A putative mirror representation system encompassing Broca's area appears to play a critical role in imitation learning and the production and perception of speech sounds[71–73].

Analysis of resting-state functional connectivity in the three participant groups provided additional information regarding the dynamics of activation in the neural network engaged by the training phase, irrespective of the particular vowel contrast to which the neonates developed sensitivity. A two-way interaction showed increased neural synchronization in both the experimental and the active control group compared with the passive control group after T1. Although the effect was quite broad, it involved strengthened connections between IF and ST regions encompassing Broca's and Wernicke's area (ST gyrus and IP lobule) in adults. Given that this pattern was found when contrasting phases T2 and T1, the increase in functional connectivity appears to have been contingent upon consolidation after initial exposure, similar to the effect on [HbO] amplitude. Supplementary to the amplitude and latency results, the neural network hinted by the functional connectivity analysis could be a neurophysiological equivalent of the sensory-motor loop theoretically linking perceptual representations of speech to motor ones during language development[74]. This offers a possible explanation for why exposure to a forward–backward vowel contrast elicited a functional connectivity increase in both the experimental and the active control groups. Such a hypothetical sensory-motor loop would tend to respond to exposure to any vocal sound that can be articulated and is thought to be crucially involved in phonological development once the motor system is functional (babbling)[75].

Vocal imitation is critical in the development of infant speech perception as it helps build synaptic projections between sensory and motor regions (that is, sensorimotor learning)[76,77]. Although vocal

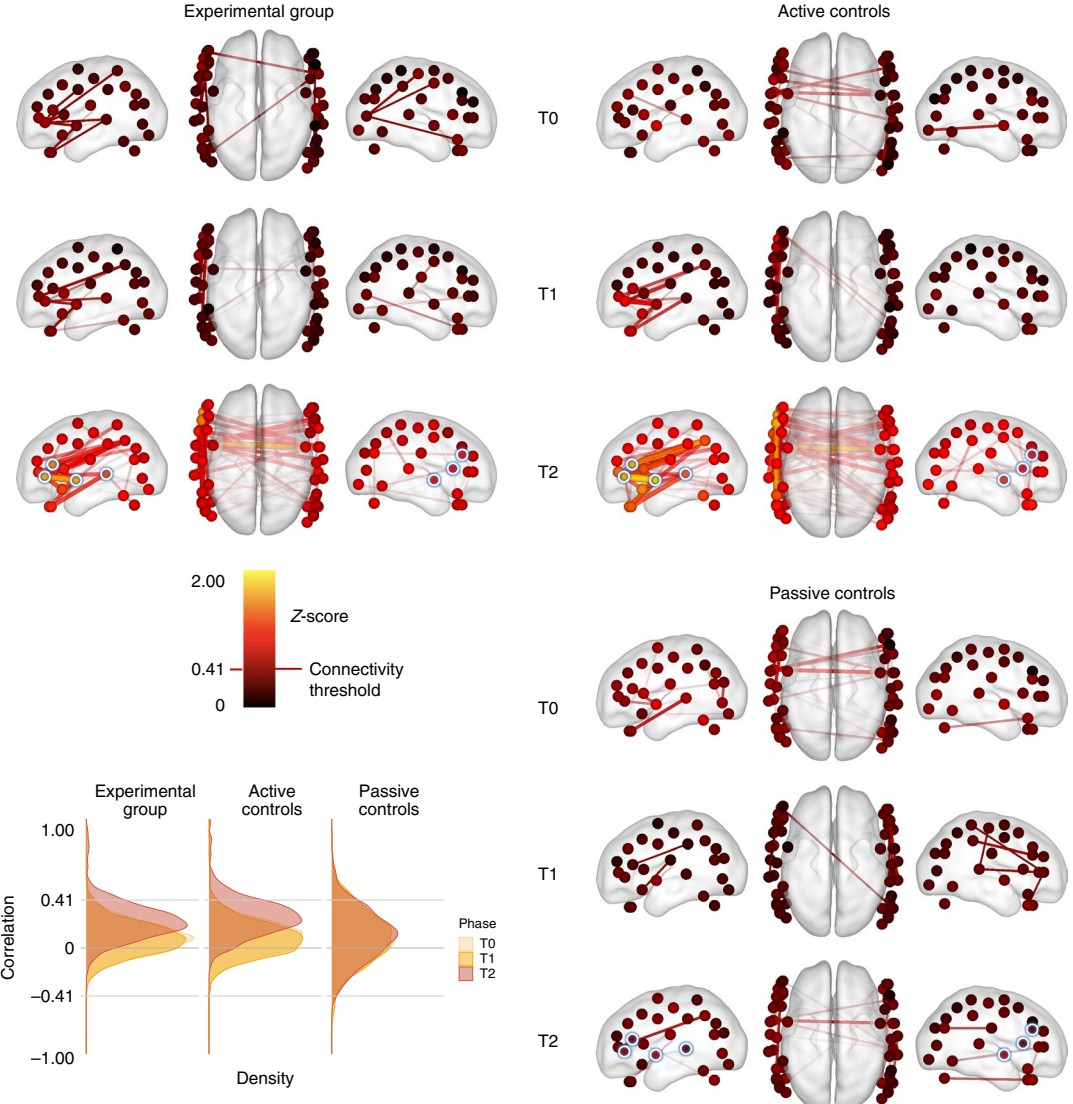

**Fig. 4 | Functional connectivity results.** Dots represent channel locations reconstructed by computing the midpoint between optodes and sensors on the basis of 3D coordinates registered for each neonate participant (Supplementary Table 3). Seed channels are highlighted with a white circle and blue halo. Dots are coloured on the basis of the average Z-score observed for the corresponding channel across the entire correlation matrix (all channels included). Lines represents correlation z-scores between pairs of channels over a threshold of 0.413, which is the absolute value of the most negative correlation observed in the experimental group at rest (see Methods and density plots in lower left quadrant). Functional connectivity intensity as measured by z-scores is depicted by hue (see colour scale), thickness (the greater the thicker) and transparency (the weaker the more transparent). The neonate brain model is from ref. [94] and visualization was implemented using the BrainNet Viewer toolbox[95].

imitation is typically observed at 6 months[78,79] (but see imitation of vowels at 20 weeks[76]), it has been shown that neonates' crying melody is affected by the language their mother speaks, suggesting possible early attempts at vocal imitation despite the anatomical limitations of the neonatal vocal tract[64]. Studies on preverbal infants have also implicated the motor system in early perceptual learning of speech[57,80]. More strikingly, constraining tongue movement has been shown to impede infants' perception of non-native speech sounds[81]. However, the developmental basis for vocal imitation remains mostly unknown. Recently, Kuhl[82] proposed that innate neural connectivity between sensory, motor and somatosensory brain areas equips infants with an experience-driven sensorimotor learning device allowing learning to begin immediately after birth (cf. the analysis by synthesis model)[83,84]. When infants babble, multimodal information is shared among the nascent language network to form an auditory-articulatory schema that supports perceiving their own

speech and, more importantly, helps infants predict the consequences of speech movements. When infants are able to imitate speech sounds they hear produced by others, their auditory-articulatory schema becomes increasingly sophisticated as a result of that experience and starts to differentiate sounds they encounter more often from those they rarely perceive (for example, native vs non-native sounds[57]). Although learning and plasticity are involved, at the centre of this account is the assumption that neural connectivity between sensory, motor and somatosensory brain areas is innate, allowing integration of sensory and motor speech processing from the very beginning of an infant's life. To our knowledge, the neural network revealed in the present study by amplitude and latency modulation, and highlighted by functional connectivity increase, provides direct evidence for the operation of such neural connectivity at birth.

In conclusion, given that we found sensitization to a subtle vowel contrast merely 5 h after exposure on the first day ex utero, neonates

**Table 3 | Estimates from the linear mixed effects regression analyses of correlation coefficients between channel pairs**

| | β | s.e.m. | d.f. | t | P |
|---|---|---|---|---|---|
| (Intercept) | 0.173 | 0.010 | 112.81 | 17.84 | <0.001 |
| Group (contrast 1: passive control vs mean (active control, experimental)) | 0.055 | 0.022 | 50.13 | 2.55 | 0.014 |
| Group (contrast 2: active control vs experimental) | −0.022 | 0.017 | 48.99 | −1.31 | 0.203 |
| Phase (contrast 1: T0 vs mean (T1, T2)) | 0.091 | 0.018 | 51.49 | 5.21 | <0.001 |
| Phase (contrast 2: T1 vs T2) | 0.182 | 0.023 | 55.29 | 7.93 | <0.001 |
| Group (contrast 1) × phase (contrast 1) | 0.098 | 0.048 | 49.01 | 2.04 | 0.047 |
| **Group (contrast 1) × phase (contrast 2)** | **0.217** | **0.062** | **50.24** | **3.48** | **0.001** |
| Group (contrast 2) × phase (contrast 1) | −0.015 | 0.038 | 51.13 | −0.39 | 0.703 |
| Group (contrast 2) × phase (contrast 2) | −0.017 | 0.048 | 49.11 | −0.36 | 0.721 |

are capable of ultra-fast tuning to natural phonemes. Nevertheless, our result remains compatible with phonological contrast tuning also taking place in the womb, depending on the particular acoustic features considered and their context of presentation[4,5,16]. Such in utero learning probably provides important bases for rapid ex utero tuning. Our study also started to unveil the neural mechanisms underpinning rapid postnatal development of phoneme perception and more specifically vowel discrimination. We provide evidence for the activation of, and increase in functional connectivity, between inferior frontal and superior temporal regions of the neonatal brain, a network reminiscent of that involved in vocal imitation in later stages of development. Future studies are needed to examine how this neural network (1) may serve as a foundation for sensorimotor learning and (2) offers an early developmental pathway to perceptual narrowing or attunement[13–15,28,76,85]. Delayed cooing and babbling are key symptoms of neurodevelopmental disorders such as autism spectrum disorder and attention deficit hyperactivity disorder[86]. By tracking the neural dynamics associated with speech sound acquisition, we may be able to better characterize newborns at risk of neurodevelopmental disorders. Further studies are required to understand how neural specialization (for example, left lateralization) gradually transforms a primordial speech acquisition network into a fully operational speech perception and production system in later life.

## Methods

**Participants.** The research was approved by the Ethical Committee of Peking University First Hospital. Seventy-five healthy full-term neonates (38 boys; gestational age: 38–41 weeks, mean = 39.0 ± 0.7 weeks) were randomly assigned to an experimental group (n = 25), an active control group (n = 25) and a passive control group (n = 25) within 1–3 h (mean = 2.1 ± 0.4 h) of birth. No statistical methods were used to pre-determine sample sizes, but our sample sizes are similar to those reported in previous publications[9,10,23,24,68]. All participants met the following criteria: (1) normal birth weight for gestational age; (2) no clinical symptoms at the time of fNIRS recording; (3) no sedation or medication before fNIRS recording; (4) normal hearing results in an evoked otoacoustic emissions test (ILO88 Dpi, Otodynamics); (5) Apgar scores higher than 8, 1 min and 5 min after birth; and (6) no neurologic abnormalities within 6 months following participation. Participants did not have any of the following neurological or metabolic disorders: (1) hypoxic-ischaemic encephalopathy, (2) intraventricular

haemorrhage or white matter damage as revealed by cranial ultrasound, (3) major congenital malformation, (4) central nervous system infection, (5) metabolic disorder, (6) clinical evidence of seizures and (7) evidence of asphyxia. Written consent was obtained from parents or the legal guardian of all participating neonates to approve access to clinical information and collection of fNIRS data for scientific purpose before data collection. Parents received ¥200 (approximately US$31.5) at the end of the experiment.

**Stimuli.** Six native vowels and their allophones (that is, /ɑː/, /ɔː/, /iː/, /uː/, /əː/ and /æ/) of standard Chinese[87] that are also common to most human languages were recorded by an adult woman, who was a native speaker of Chinese with the Peking dialect. A single token was recorded for each vowel, which was then edited to have a duration of 1 s using Cool Edit Pro 2.1 (Syntrillium software). A short silence was added to make each sound file 1 s long and no other modification (that is, compression or expansion) was applied. Pitch contours and spectrograms of the 6 vowels used are shown in Supplementary Fig. 1. Note that the duration of the vowel sound '/æ/' was shorter than that of the other vowels. However, given that the critical manipulation contrasted vowels and their temporally reversed counterpart, forward and backward stimuli had the exact same duration. In addition, previous studies have shown that length difference is not a salient factor for phonemic perception in neonates[18]. As can be seen in Supplementary Fig. 1, the stimuli contained minimal temporal variations in their fundamental frequencies, suggesting absence of prosodic information. A group of 20 Chinese undergraduate students (10 males, mean age = 21.4 years, s.d. = 1.7) with the same language background as the participating neonates' parents performed a recognition task and a prosodic rating task on the stimuli used in the experiment. In the recognition task, 12 stimuli (that is, 6 vowels and their backward versions) were randomly presented to the participants who were asked to match the order of their presentation to written sequences provided on a leaflet. A 'condition' (forward vs backward) by 'group' (active control vs experimental) two-way ANOVA showed a main effect of condition, $F(1,19) = 21.92$, $P < 0.001$, $\eta_p^2 = 0.536$, where the recognition accuracy was much higher for forward vowels (M = 98.3%, s.d. = 10.6) as compared with backward vowels (M = 73.2%, s.d. = 33.2). In the rating task, participants rated prosodic variations of the 12 stimuli on a 9-point scale (1 being the least prosodic and 9 being the most prosodic). A two-way ANOVA showed no significant effect either of 'condition' or 'group', and prosodic ratings were very low in all conditions (all means <1.2, s.d. = 0.20).

In the experimental group, we used 12 naturally pronounced (forward) vowel strings, each containing 6 concatenated vowels (that is, /ɑː/, /ɔː/ and /iː/ repeated twice; Supplementary Table 2). The non-speech sound included the same 12 vowels played backwards[24,63]. Forward sounds used in the active control group during the learning phase comprised 12 forward vowel strings, each containing 6 concatenated vowels (that is, /uː/, /əː/ and /æ/ repeated twice; Supplementary Table 2). As in the case of the experimental group, backward sounds used in the active control group were the same 12 vowels played backwards. The presentation order of the backward vowels always matched that of the forward vowels. Furthermore, forward and backward stimuli were matched in terms of frequency range and intensity between training phases in the experimental and the control groups.

**Procedure.** The experiment was conducted in the neonatal ward of Peking University First Hospital, Beijing, China. Neonates were transported to a dedicated testing room as soon as their state was stable after birth. There, they were separated from their mothers to minimize natural exposure to speech or speech stimuli other than those used in the experiment. The auditory stimuli were presented through a pair of loudspeakers placed 20 cm away from the neonates' left and right ears, at a sound pressure level of 55 to 60 dB. The mean background noise intensity level was 30 dB. NIRS recording was carried out when the neonates were in a quiet state of alert or in a state of natural sleep[23]. Neonates who cried for more than 2 min during the recording were excluded from analyses, and this left 22 (11 boys), 23 (12 boys) and 21 (10 boys) datasets to be included in the experimental, active control and passive control groups, respectively.

At the beginning of the experiment, a baseline recording of 8 min (T0) was performed during which forward trials (that is, 12 forward vowel strings) and backward trials (that is, 12 backward vowel strings) were randomly presented. Each trial consisted of 6 vowels (or backward vowels) and had a duration of 6 s (Supplementary Table 2, left column). Inter-trial intervals were silent and varied randomly in duration between 12 and 16 s. Immediately after the baseline test, the training phase started during which neonates in the experimental group were presented with experimental forward and backward vowel sets in blocks (Fig. 1). The 'speech' block featuring natural vowels comprised 72 trials of forward vowel strings (6 repetitions of the 12 experimental stimuli), with a fixed 2 s inter-trial interval. The 'non-speech' block had the same structure but comprised backward vowel strings. Inter-block interval was 24 s. Each block lasted for 10 min, and the order of forward and backward blocks was counterbalanced across participants. The forward and backward blocks were each repeated 15 times, resulting in a 30-block training phase lasting 5 h. The active control group received the same training, albeit with vowels different from those used for testing.

A test (T1) involving the same procedure as the baseline (that is, 8 min random presentation of forward and backward sounds) was conducted 5 min after the end

of the training phase. After T1, the neonates slept for 2 h (that is, a consolidation period) before they were tested again (T2) using the same procedure as in T0 and T1. All three groups of participants underwent identical data collection at T0, T1 and T2. The passive control group did not receive any training but was also placed in the same testing room following the same procedure as the other two groups of participants. The experiment lasted for approximately 7.5 h (Fig. 1) and was conducted in a blind fashion since the nurses who assisted with data collection were only briefed about the purpose of the study after data collection had ended.

Neonate participant state was continuously monitored using a video monitoring system and a polysomnography device (Ventmed Medial Technology). Their eye movements, body movements, heart rate and respiration rate were collected. Results showed that neonates were asleep most of the time (>90%) during the consolidation period between T1 and T2 and there was no significant difference between groups (P > 0.3; Supplementary Table 4).

**NIRS data recording.** NIRS data was collected at T0, T1 and T2 in continuous-wave mode using the NirSmart system (Danyang Huichuang), consisting of 20 laser emitters (mean intensity = 2 mW per wavelength) and 16 optodes (light detectors) sensitive to 2 wavelengths (760 and 850 nm). The optodes were distributed evenly across the scalp, covering the temporal and frontal regions in particular, and were set in a NIRS-EEG compatible 34-cm-diameter cap (EASYCAP) in accordance with the international 10/5 system. There were 52 channels (symmetrical between hemispheres, Supplementary Fig. 2), with source and detectors set at a mean distance of 2.3 cm. Distances between the source and the detector for each channel are listed in Supplementary Table 3. The data were recorded continuously at a sampling rate of 10 Hz.

**Data pre-processing.** Pre-processing and statistical analyses were conducted in Matlab (v2021a, Mathworks). NIRS data were first screened manually to check for detector saturation, which occurred in none of the participants and none of the channels. Data epochs containing large artefact (>20% dynamic range of the device input) were removed (17.8 ± 10.2 % of data was removed in this step). After automatic detection (peak-to-peak >6 s.d.) and correction of spike 'jumps' using linear interpolation, optical intensity data were converted into optical density variations (ΔOD), followed by band-pass filtering between 0.01–0.2 Hz[25,59]. Filtered ΔOD timeseries for both wavelengths of interest were then transformed into relative concentration changes of oxyhaemoglobin (Δ[HbO]) and deoxyhemoglobin (Δ[Hb]), respectively. In this study, data distribution was assumed to be normal, but this was not formally tested.

**Variations in oxyhemoglobin concentration between conditions.** The classical general linear model (GLM) assumes that the haemodynamic response function (HRF) in a given brain region is time-invariant within a participant, which might not always be true in infants[88]. Indeed, the observed Δ[HbO] waveforms displayed variable within-subject HRF timecourses in this study. Thus, we chose not to use an HRF-based GLM approach in our statistical analyses[9,23,24,63,68] and resorted to comparing simple measures in Δ[HbO] and Δ[Hb] waveforms instead, that is, mean amplitude and peak latency[9,24,68]. Continuous Δ[HbO] and Δ[Hb] data were segmented starting at 2 s before the onset of a stimulus and until 20 s after. Epochs were baseline-corrected with respect to pre-stimulus mean concentration. Although both Δ[HbO] and Δ[Hb] were computed, we focused on Δ[HbO] since it best reflects neural activation[9,63]. Our primary analysis first calculated mean amplitude of the Δ[HbO] value from 6–16 s after stimulus onset in each trial. This time window was expected to contain the maximum changes in stimulus-related oxygenated haemoglobin concentration on the basis of previous literature[9,24,31,68]. Then, peak latencies of the Δ[HbO] waveforms were detected automatically (maximum over the whole epoch duration) in each trial.

Δ[HbO] mean amplitude and peak latency were modelled using linear mixed effects regression, via the R package lme4[89]. Single-trial amplitudes or latencies were modelled as a function of three centred, sum-coded fixed effects (stimulus type, participant group and test phase) and their interactions. Stimulus type was coded as a binomial contrast (backward vowels, forward vowels). Participant group was coded as a three-level Helmert contrast: the first constituent contrast encoded the general effect of training by comparing the passive control group to the mean of the active control group and the experimental group; the second constituent contrast encoded the specific effect of training on relevant content by comparing the active control group to the experimental group. Testing phase was also coded as a three-level Helmert contrast: the first constituent contrast encoded the general effect of time, relative to training, by comparing the pre-training baseline (T0) to the mean of the two post-training tests (T1 and T2); the second constituent contrast encoded the more specific effect of sleep and consolidation by comparing the test immediately after training (T1) to the test 2 h later (T2). All predictors were mean-centred and all models included maximal by-participant and by-channel random effects structures, omitting random effect correlations to facilitate convergence[90]; by-participant random effects structures necessarily omitted between-participants contrasts and by-channel random effects structures were necessarily omitted for the individual channel analyses. P-value calculations used the Satterthwaite approximation method, implemented via the lmerTest package[91]. Models provided beta estimates, in the form of BLUPs, for each level of the random

grouping variables (per channel for the amplitude and latency analyses; per channel pair for the connectivity analysis); where appropriate, these estimates were supplemented by refitting the same model to restricted datasets for each level.

Resting-state connectivity between brain regions was estimated by correlating the 10 Hz measurements between pairs of detectors for each subject 3 min (1,800 samples) before each test[35]. After applying a band-pass filter (0.01–0.2 Hz), ΔOD signals were converted to Δ[HbO][35,74]. We then calculated Pearson correlations between timeseries for each pair of fNIRS channels to estimate spontaneous functional connectivity[35], producing matrices of correlation coefficients (r values). The r values were then transformed to Fisher's z-scores for further statistical analyses, to approximate a normal distribution[35] (that is, $z = \operatorname{artanh}(r)$). To assess changes in the strength of these connections as a function of experience, linear mixed effects regressions then fitted reduced forms of the same models that were used for the amplitude and latency analyses, including maximal random effects structures for each subject and each pair of channels. To ensure that the results of the regression analysis would reflect changes that were relevant to the amplitude and latency effects that were our main focus, we included only the pairwise correlations involving 7 seed channels that had shown the crucial amplitude and latency effects after FDR correction (FDR threshold: 0.15, yielding channels 7, 10 and 45 from the amplitude analysis, and channels 2, 6, 43 and 44 from the latency analysis). Thus, we calculated 336 Pearson correlations between timeseries from seed channels and all other channels ($7 \times 51 - (7 \times 6)/2 = 336$). The regression models necessarily omitted the fixed and random effects of stimulus type and its interactions because the pre-test period did not include stimuli of either type. They retained the second group contrast (active control vs experimental) as structural consideration, although no valid main effects or interactions for this predictor were anticipated in this analysis because these active groups differed only in the match between their training and testing stimuli.

**Spatial registration for NIRS channels.** We first measured the distance between participants' nose concave point and head back bulge point to locate channel Cz, which was then marked with a white dot on the cap. We used Cz, Nz (above the nose concave point), Iz (the inion bulge point), AR (above the right ear) and AL (above the left ear) as references to position the cap on the participant's head. The spatial coordinates of these 5 channels were matched onto locations in the neonatal head model[92] using a three-dimensional (3D) digitizer (Patriot), which then registered the locations of 20 sources and 16 detectors distributed over the neonate's head. NIRS channel locations were defined as the central zone on the path the light travels between each adjacent source–detector pair. After optode registration, NIRS sources and detectors were put on the cap and prepared for data recording. Coordinates were subsequently averaged across participants. Using reference positions as a guide, the channel coordinates were projected onto the cortical surface of a neonatal MNI cortex model[93]. The Montreal Neurological Institute (MNI) coordinates were then mapped onto a neonatal Automated anatomical labelling (AAL) brain atlas[92]. Results of the channel registration are listed in Supplementary Table 3.

**Reporting summary.** Further information on research design is available in the Nature Research Reporting summary linked to this article.

## Data availability
The data for linear mixed effects regression as well as a full report of the NIRS results are uploaded as Supplementary Information. Additional data would be available upon reasonable request and with approval of the School of Psychology, Shenzhen University. More information on making this request can be obtained from D.Z. (zhangdd05@gmail.com).

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

## Acknowledgements

This study was funded by the National Natural Science Foundation of China (31970980; 31920103009), the National Social Science Fund of China (21AYY014), National Key Research and Development Program of China (2021YFC2700700), Beijing Municipal Science and Technology Commission (Z19110000661904), Shenzhen-Hong Kong Institute of Brain Science (2021SHIBS0003) and the Fundamental Research Funds for the Provincial Universities of Zhejiang. G.T. is supported by the Polish National Agency for Academic Exchange (NAWA) under the NAWA Chair Programme (PPN/PRO/2020/1/00006). The funders had no role in study design, data collection and analysis, decision to publish or preparation of the manuscript.

## Author contributions

Y.J.W., X.H. and D.Z. conceived and designed the experiment. X.H., C.P. and W.Y. collected the data. G.M.O., D.Z. and G.T. analysed the data. Y.J.W. and D.Z. wrote the first draft of the paper. G.T. created the figures. Y.J.W., D.Z., G.T. and G.M.O. revised the paper.

## Competing interests

The authors declare no competing interests.

## Additional information

**Correspondence and requests for materials** should be addressed to Dandan Zhang.

# Reporting Summary

## Statistics

For all statistical analyses, confirm that the following items are present in the figure legend, table legend, main text, or Methods section.

| n/a | Confirmed | |
|---|---|---|
| ☐ | ☒ | The exact sample size (*n*) for each experimental group/condition, given as a discrete number and unit of measurement |
| ☐ | ☒ | A statement on whether measurements were taken from distinct samples or whether the same sample was measured repeatedly |
| ☐ | ☒ | The statistical test(s) used AND whether they are one- or two-sided *Only common tests should be described solely by name; describe more complex techniques in the Methods section.* |
| ☒ | ☐ | A description of all covariates tested |
| ☐ | ☒ | A description of any assumptions or corrections, such as tests of normality and adjustment for multiple comparisons |
| ☐ | ☒ | A full description of the statistical parameters including central tendency (e.g. means) or other basic estimates (e.g. regression coefficient) AND variation (e.g. standard deviation) or associated estimates of uncertainty (e.g. confidence intervals) |
| ☐ | ☒ | For null hypothesis testing, the test statistic (e.g. *F*, *t*, *r*) with confidence intervals, effect sizes, degrees of freedom and *P* value noted *Give P values as exact values whenever suitable.* |
| ☒ | ☐ | For Bayesian analysis, information on the choice of priors and Markov chain Monte Carlo settings |
| ☒ | ☐ | For hierarchical and complex designs, identification of the appropriate level for tests and full reporting of outcomes |
| ☐ | ☒ | Estimates of effect sizes (e.g. Cohen's *d*, Pearson's *r*), indicating how they were calculated |

*Our web collection on statistics for biologists contains articles on many of the points above.*

## Software and code

Policy information about availability of computer code

| Data collection | NIRS Data was collected using the NirSmart system (Danyang Huichuang, China). Sound materials were edited using Cool Edit Pro 2.1 (Syntrillium Software Corp., AZ, USA). |
|---|---|
| Data analysis | Pre-processing and statistical analyses were conducted in Matlab (v2021a, the Mathworks, Inc., Natick, USA). NIRS Data analysis was performed using the NirSmart system (Danyang Huichuang, China). Linear mixed effects models were conducted using the R package lme4 (Bates et al., 2015). |

For manuscripts utilizing custom algorithms or software that are central to the research but not yet described in published literature, software must be made available to editors and reviewers. We strongly encourage code deposition in a community repository (e.g. GitHub). See the Nature Portfolio guidelines for submitting code & software for further information.

## Data

Policy information about availability of data

All manuscripts must include a data availability statement. This statement should provide the following information, where applicable:
- Accession codes, unique identifiers, or web links for publicly available datasets
- A description of any restrictions on data availability
- For clinical datasets or third party data, please ensure that the statement adheres to our policy

The data for linear mixed effects regression as well as a full report of the NIRS results are upload as supplementary files. Additional data would be available upon reasonable request and with approval of the School of Psychology, Shenzhen University. More information on making this request can be obtained from the corresponding author, D. Zhang (zhangdd05@gmail.com).

# Field-specific reporting

Please select the one below that is the best fit for your research. If you are not sure, read the appropriate sections before making your selection.

☒ Life sciences ☐ Behavioural & social sciences ☐ Ecological, evolutionary & environmental sciences

For a reference copy of the document with all sections, see nature.com/documents/nr-reporting-summary-flat.pdf

# Life sciences study design

All studies must disclose on these points even when the disclosure is negative.

| | |
|---|---|
| Sample size | Seventy-five healthy full-term neonates were randomly assigned into the experimental group (n = 25), the active control group (n = 25), and the passive control group (n = 25). Since data collection in healthy, full-term neonates has practical limits, the sample size was based on previous studies on neonates with similar research objectives (e.g., Benavides-Varela et al., PNAS, 2012; Cabrera & Gervain, Sci Adv, 2020; Gervain et al., PNAS, 2008; Gómez et al., PNAS, 2014; May et al., Dev Sci, 2018; Peña et al., PNAS, 2003; Perani et al., PNAS, 2011). |
| Data exclusions | Neonates who started crying during the recording were excluded from the analyses (i.e., 3 from the experimental group, 2 from the active control group, and 4 from the passive control group). |
| Replication | Results in the current study are not replicated because the study was conducted on neonates which requires a substantial period of time to collect data and, therefore, to replicate the current study. However, details of the experiments (i.e.,procedure and stimuli) and data analysis are provided in the manuscript, allowing future replications of this study. |
| Randomization | Participants were randomly assigned into the experimental and control groups. |
| Blinding | The investigator who collected the data was blinded to group allocation during data collection and was debriefed with the purpose of the study afterwards. The researcher who analyzed the data was also blind to the conditions of the experiment, as the conditions were coded during data analysis. |

# Reporting for specific materials, systems and methods

We require information from authors about some types of materials, experimental systems and methods used in many studies. Here, indicate whether each material, system or method listed is relevant to your study. If you are not sure if a list item applies to your research, read the appropriate section before selecting a response.

## Materials & experimental systems

| n/a | Involved in the study |
|---|---|
| ☒ | Antibodies |
| ☒ | Eukaryotic cell lines |
| ☒ | Palaeontology and archaeology |
| ☒ | Animals and other organisms |
| ☐ | ☒ Human research participants |
| ☒ | Clinical data |
| ☒ | Dual use research of concern |

## Methods

| n/a | Involved in the study |
|---|---|
| ☒ | ChIP-seq |
| ☒ | Flow cytometry |
| ☒ | MRI-based neuroimaging |

# Human research participants

Policy information about studies involving human research participants

| | |
|---|---|
| Population characteristics | Seventy-five healthy, full-term neonates (38 boys; gestational age: 38 to 41 weeks, mean = 39.0 ± 0.7 weeks) within 1 to 3 hours (mean = 2.1 ± 0.4 hours) of birth were recruited in the study. |
| Recruitment | Recruitment advertisement was posted at the entrance to the obstetrics department of Peking University First Hospital. Parents who were willing to take part in the study contacted the experimenter and their information was recorded. Among these recorded neonates, the ones who qualified the requirements of the study (see Participants in the Methods section) were recruited on the first day after birth. Written consent was obtained from parents prior to data collection. |
| Ethics oversight | This study was approved by the Ethical Committee of Peking University First Hospital. |

Note that full information on the approval of the study protocol must also be provided in the manuscript.

