## [Peer Review File · Nature Human Behaviour]

Peer Review Information

Journal: Nature Human Behaviour

Manuscript Title: Rapid learning of a phonemic discrimination in the first hours 2 of life

Corresponding author name(s): Dandan Zhang

Reviewer Comments & Decisions:

Decision Letter, initial version:
--

23rd September 2021

Dear Professor Zhang,

Thank you once again for your manuscript, entitled "Sensory-motor neural network supports ultra-fast tuning to natural vowels in neonates", and for your patience during the peer review process.

Your Article has now been evaluated by 2 referees. You will see from their comments copied below that, although they find your work of considerable potential interest, the referees have raised quite substantial concerns. In light of these comments, we cannot accept the manuscript for publication, but would be interested in considering a revised version if you are willing and able to fully address reviewer and editorial concerns.

We hope you will find the referees' comments useful as you decide how to proceed. If you wish to submit a substantially revised manuscript, please bear in mind that we will be reluctant to approach the referees again in the absence of major revisions. We are committed to providing a fair and constructive peer-review process. Do not hesitate to contact us if there are specific requests from the reviewers that you believe are technically impossible or unlikely to yield a meaningful outcome.

In particular, Referee #2 raises a number of concerns about the fNIRS results, suggesting new analyses and questioning the credibility of some of the results namely the latency analysis. A revised version of the manuscript would need to fully address all of these concerns. Please note that moving the latency analyses to the Supplementary Information would not be sufficient - these analyses must either be corrected with re-analysis, or removed.

Finally, your revised manuscript must comply fully with our editorial policies and formatting requirements. Failure to do so will result in your manuscript being returned to you, which will delay its consideration. To assist you in this process, I have attached a checklist that lists all of our

requirements. If you have any questions about any of our policies or formatting, please don't hesitate to contact me.

If you wish to submit a suitably revised manuscript we would hope to receive it within 6 months. We understand that the COVID-19 pandemic is causing significant disruptions which may prevent you from carrying out the additional work required for resubmission of your manuscript within this timeframe. If you are unable to submit your revised manuscript within 6 months, please let us know. We will be happy to extend the submission date to enable you to complete your work on the revision.

- Include a "Response to the editors and reviewers" document detailing, point-by-point, how you addressed each editor and referee comment. If no action was taken to address a point, you must provide a compelling argument. This response will be used by the editors to evaluate your revision and sent back to the reviewers along with the revised manuscript.
- Highlight all changes made to your manuscript or provide us with a version that tracks changes.

[REDACTED]

Thank you for the opportunity to review your work. Please do not hesitate to contact me if you have any questions or would like to discuss the required revisions further.

Sincerely,
Jamie

Dr Jamie Horder
Senior Editor
Nature Human Behaviour

Reviewer expertise:

Reviewer #1: infant language and auditory perception

Reviewer #2: infant fNIRS and perception

REVIEWER COMMENTS:

Reviewer #1:

Remarks to the Author:

Review of: Sensory-motor neural network supports ultra-fast tuning to natural vowels in neonates

Overview:

The authors provide data suggesting that neonates show differential and very rapid learning of forward vowels as opposed to backward vowels after birth. Using fNIRS, the authors test infants' brain activation to vowels prior to any speech experience (baseline). The infants are then randomly assigned to experimental (listen to forward and backward vowels for 5 hours), active controls (listen to a different set of forward and backward vowels for 5 hours) and controls (no listening experience), and they repeat the fNIRS measure immediately after the 5-hour listening period (Time 1), and then again two hours later (Time 2). Results showed that at T1, E participants had faster hemodynamic responses in the sensors above inferior frontal for natural than reverse vowels. At T2, increased response was seen in the STG and left inferior parietal. Neither of the Control groups showed this response. Stronger functional connectivity between sensory and motor areas was also observed for natural as opposed to reversed vowels, but not by Controls.

The data are interesting and important, but the authors' message is obscured in my opinion by unnecessary claims about (1) whether their data address assumptions that infants have universal abilities to discriminate vowels at birth (these data cannot determine the veracity of that claim), (2) for lacking care in describing what they have tested by referring to 'backward-played' vowels as 'nonspeech' when their own data on adults show that their vowels played backwards were correctly identified 73% of the time, and (3) for referring to their fNIRS sensor data as measuring brain activation in a particular place (e.g., the inferior frontal gyrus) rather than more carefully stating that the sensors placed above what is assumed to be the inferior frontal area on a neonate's head were activated. Some brain activation data, such as MEG, can be used to calculate the source of brain activation, but that is not true of fNIRS data. This does not invalidate the importance of the data, but scholarly accuracy needs to be preserved. All of these things reflect a lack of sophistication, but can and should be dealt with in a revision. There are citations that should also be included for scholarly completeness.

General Issues

The three issues mentioned above are the most important:

(1) The authors rebuttal of the largely held belief that newborns at birth can discriminate vowels, is distracting, and only tangentially supported by their data (because they used vowels played forward and backward with virtually no pitch changes making them more like singing than speaking, which makes them not so non-speech like as claimed). More importantly, it is unnecessary. The value of the current data is that neonates learn, and quickly, to focus on the forward vowels, over backwards vowels. These learning data are very interesting. Distracting readers with a claim that's weak takes away, rather than adds, to the narrative.

(2) The authors use simple vowels without prosodic variation – they are more like sung than spoken vowels. This means they do not resemble nonspeech as much as the authors claim. The authors should not refer to these stimuli as ‘nonspeech’ but instead repeat in each case, and on all figures, that they were forward vs. backward vowels. The authors’ own data on adults (who were asked to identify the vowels), identified forward vowels 98% of the time correctly, as expected, but they also identified the backward vowels correctly 73% of the time. This is not nonspeech (which cannot be identified as speech). Please describe the stimuli as forward and backward vowels to keep readers straight about what’s being tested. With experience, the forward and backward vowels were clearly differentiated.

(3) The use of fNIRS is reasonable for neonates, but the authors should be conservative about source localization in the brain of the neonates. The measurements are taken from the sensors on the infants’ heads, and the sensors are placed where the authors believe important neural structures are located, but this is approximate in a tiny neonate’s head. And, the sensors on the baby’s head are not picking up only brain activity from one brain structure, such as the inferior frontal, but activation from other nearby areas – sensors are activated by ‘bleeding’ of activation from one brain area to another. For fNIRS ‘source’ localization is nascent. So the authors should refer to the results as derived from fNIRS sensors located approximately above inferior frontal areas, so readers remembers that this is ‘sensor’ data, not ‘source’ estimate data.

(4) There are references that would aid the authors’ arguments. One is Imada et al. (2006, NeuroReport) in which neonates, 6-month-olds, and 12-month-olds listened to speech syllables and two nonspeech stimuli, pure tones, and harmonics that resemble aspects of vowels but cannot be identified as speech, using MEG technology. Neonates did not show activation in IF areas for any stimuli, while showing STG activation to all stimuli. But by 6 months, and again at 12 months, speech stimuli showed timed activation in IF and STG, and this was not shown by the nonspeech stimuli. This should be referenced on lines 403, 404. Kuhl et al (2014, PNAS) is referenced, but should also be cited on lines 407,408. A new MEG study, using syllables to test infants, again identifies IF as the critical area for predicting future language ability (Zhao et al., 2021, Developmental Cognitive Neuroscience, doi.org/10.1016/j.dcn.2021.100949. Finally there is a chapter by Kuhl in 2021, Minnesota symposia on child psychology: Human communication: Origins, mechanism, and functions (Vol. 40, pp. 113–149), published by Wiley, that describes a new view regarding the sensorimotor bases of infant speech perception that the authors might find helpful.

Reviewer #2:

Remarks to the Author:

This manuscript reports on a heroic study of human newborns in their first postnatal day. They were presented with one of three speech exposure conditions for a total of 5 hours, with a pre-test, an immediate post-test, and a follow-up post-test 2 hours later. At issue is whether newborns’ vowel discrimination is affected by the types of vowels to which they are exposed in their first day after birth. Results suggest that they are, indeed, affected by this early speech exposure. As such, these results represent a quite impressive example of early auditory learning. And if true, they should certainly be published in a high-impact journal like Nature Human Behavior.

Below I make some suggestions for improvement in the manuscript, as well as some reservations about both the methods and interpretation.

First, the overall design of this study is based on Cheour et al. (2002) who showed that basic vowel discrimination can be obtained using EEG with newborns. And, in fact, there are many such demonstrations of vowel discrimination in early infancy. So the Introduction should not dwell on some failures of vowel discrimination which can be chalked up to poor methods or very subtle vowel distinctions (i.e., with low-salience acoustic cues).

Second, the specific choice of vowel discrimination in the present study is quite unusual. The authors say they took great pains to use steady-state vowels (i.e., not vowels in consonant contexts) and to use only one token of each of 12 vowels. The vowel contrast, then, was not about a robust vowel category difference in Chinese, but rather a VERY subtle contrast between a forward vowel and its waveform reversal. It is not clear why the authors chose this contrast because steady-state vowels, by definition, have the SAME spectral information whether forward or reversed (and no prosodic differences as verified by adult ratings). However, upon closer examination (see Supplemental Fig. 4), there were two acoustic cues that differentiated the forward vs. reversed vowels – (1) a rapid vowel-onset and amplitude increase and a gradual vowel-offset and amplitude decrease in the forward tokens vs. the opposite pattern in the reverse tokens, and (2) the presence of vowel-onset glottalization in the forward tokens and vowel-offset glottalization in the reverse tokens. These two differences in the reverse tokens are not “natural” (as reflected in the ratings by adults), but that is not particularly relevant in the present design because prenatal acoustic exposure is unlikely to have provided the fetuses with these cues due to the low-pass filtering of the uterus. The main point here is that these are VERY subtle differences and therefore would be expected to be difficult to discriminate by newborns.

Third, the fact that only the experimental condition showed a learning effect after 5 hours of exposure implies that the active control group, who also listened to 5 hours of DIFFERENT vowels, were unable to extract the differences in amplitude onset/offset and glottalization onset/offset and generalize them to the test tokens. This is a remarkably NARROW example of learning (i.e., specific only to the vowels presented during the exposure phase).

Fourth, the consolidation effect in the experimental group is quite interesting, especially since many phonetic learning effects in infants fail to be retained even after a few minutes delay.

In summary, you should emphasize the preceding four points because they set the stage for why these findings are so impressive.

Fifth, fNIRS is not a terribly sensitive method for detecting subtle auditory discrimination abilities, especially in newborns. And the use of a latency measure is particularly unusual (Dehaene-Lambertz et al. 2006 used it with fMRI but that was with sentences, not isolated vowel syllables). For example, the really short latencies at T1 and T2 in the experimental group are highly suspect. I have never seen such rapid rises in the HRF in ANY fNIRS study, suggesting they are due to spontaneous background fluctuations in the underlying hemodynamics. I suggest moving these latency results to the Supplemental Materials.

Sixth, you should conduct a linear mixed model to analyze your fNIRS amplitude effects rather than a set of ANOVAs.

Seventh, you should conduct a permutation test with your functional connectivity analyses to guard against false positives. How do we know that similar connectivity results were not present in other (i.e., non-language related) ROIs?

Eighth, you need to explain in more detail how you did anatomical co-registration of the fNIRS optodes. This is crucial for analyses that involve SINGLE fNIRS channels because they must sample the same underlying cortical regions across the 25 newborns in each of the three experimental conditions.

Ninth, I found your General Discussion about possible motor involvement in these findings to be highly speculative. You would need a different experimental design to confirm that interpretation.

In conclusion, this manuscript presents some really interesting findings, and I am cautiously optimistic that the authors can deal with my concerns and produce a revision that will be acceptable for publication.

Author Rebuttal to Initial comments

Point-by-point reply to Reviewers' comments

Reviewer #1:

The data are interesting and important, but the authors' message is obscured in my opinion by unnecessary claims about (1) whether their data address assumptions that infants have universal abilities to discriminate vowels at birth (these data cannot determine the veracity of that claim), (2) for lacking care in describing what they have tested by referring to 'backward-played' vowels as 'nonspeech' when their own data on adults show that their vowels played backwards were correctly identified 73% of the time, and (3) for referring to their fNIRS sensor data as measuring brain activation in a particular place (e.g., the inferior frontal gyrus) rather than more carefully stating that the sensors placed above what is assumed to be the inferior frontal area on a neonate's head were activated. Some brain activation data, such as MEG, can be used to calculate the source of brain activation, but that is not true of fNIRS data. This does not invalidate the importance of the data, but scholarly accuracy needs to be preserved. All of these things reflect a lack of sophistication, but can and should be dealt with in a revision. There are citations that should also be included for scholarly completeness.

General Issues

The three issues mentioned above are the most important:

1. The authors rebuttal of the largely held believe that newborns at birth can discriminate vowels, is distracting, and only tangentially supported by their data (because they used vowels played forward and backward with virtually no pitch changes making them more like singing than speaking, which makes them not so non-speech like as claimed). More importantly, it is unnecessary. The value of the current data is that neonates learn, and quickly, to focus on the forward vowels, over backwards vowels. These learning data are very interesting. Distracting readers with a claim that's weak takes away, rather than adds, to the narrative.

Thank you for this suggestion. In the revised manuscript, we have removed the claim that the current study can shed light onto phonemic perception capacity at birth and we focus instead on how the current data provide mechanistic insights into rapid postnatal acquisition of vowels. [see page 4, lines 65-67; also see page 12, lines 222-225].

2. The authors use simple vowels without prosodic variation – they are more like sung than spoken vowels. This means they do not resemble nonspeech as much as the authors claim. The authors should not refer to these stimuli as 'nonspeech' but instead repeat in each case, and on all figures, that they were forward vs. backward vowels. The authors' own data on adults (who were asked to identify the vowels), identified forward vowels 98% of the time correctly, as expected, but they also identified the backward vowels correctly 73% of the time. This is not nonspeech (which cannot be identified as speech). Please describe the stimuli as forward and backward vowels to keep readers straight about what's being tested. With experience, the forward and backward vowels were clearly differentiated.

We absolutely agree with Reviewer 1 and we have implemented this change throughout the manuscript, including in all figures and tables.

3. The use of fNIRS is reasonable for neonates, but the authors should be conservative about source localization in the brain of the neonates. The measurements are taken from the sensors on the infants' heads, and the sensors are placed where the authors believe important neural structures are located, but this is approximate in a tiny neonate's head. And, the sensors on the baby's head are not picking up only brain activity from one brain structure, such as the inferior frontal, but activation from other nearby areas – sensors are activated by 'bleeding' of activation from one brain area to another. For fNIRS 'source' localization is nascent. So the authors should refer to the results as derived from fNIRS sensors located approximately above inferior frontal areas, so readers remembers that this is 'sensor' data, not 'source' estimate data.

We agree with the Reviewer that fNIRS does not provide a direct measure of neural activity from specific brain regions and also that its spatial resolution is poor. In the revised manuscript we are more cautious about the reference to neuroanatomical substrate by referring to fNIRS channels as placed approximately over certain brain regions rather than representing specific structures such as gyri (for instance, channel

6 is approximately placed over the Superior Temporal region). More importantly, our new LMER-based analytical approach shifts the focus away from individual channels, although we still report the spatial distribution of the effects.

4. There are references that would aid the authors' arguments. One is Imada et al. (2006, NeuroReport) in which neonates, 6-month-olds, and 12-month-olds listened to speech syllables and two nonspeech stimuli, pure tones, and harmonics that resemble aspects of vowels but cannot be identified as speech, using MEG technology. Neonates did not show activation in IF areas for any stimuli, while showing STG activation to all stimuli. But by 6 months, and again at 12 months, speech stimuli showed timed activation in IF and STG, and this was not shown by the nonspeech stimuli. This should be referenced on lines 403, 404. Kuhl et al (2014, PNAS) is referenced, but should also be cited on lines 407,408. A new MEG study, using syllables to test infants, again identifies IF as the critical area for predicting future language ability (Zhao et al., 2021, Developmental Cognitive Neuroscience, doi.org/10.1016/j.dcn.2021.100949). Finally there is a chapter by Kuhl in 2021, Minnesota symposia on child psychology: Human communication: Origins, mechanism, and functions (Vol. 40, pp. 113–149), published by Wiley, that describes a new view regarding the sensorimotor bases of infant speech perception that the authors might find helpful.

We are grateful for these important references. In the revised manuscript, we now cite Imada et al., (2006), Kuhl et al., (2014) and Zhao et al., (2021) as key references when discussing the involvement of IF regions in early language processing and speech development [see page 14, line 275-276; page 16, line 348-350]. We thank the Reviewer in particular for bringing the Kuhl (2021) reference to our attention, as it is indeed highly relevant. We now cite this reference in relation to our hypothesis regarding sensorimotor connections in early speech development and vocal imitations as a possible underlying mechanism [see page 16, lines 352-368].

Reviewer #2:

Remarks to the Author:

This manuscript reports on a heroic study of human newborns in their first postnatal day. They were presented with one of three speech exposure conditions for a total of 5 hours, with a pre-test, an immediate post-test, and a follow-up post-test 2 hours later. At issue is whether newborns' vowel discrimination is affected by the types of vowels to which they are exposed in their first day after birth. Results suggest that they are, indeed, affected by this early speech exposure. As such, these results represent a quite impressive example of early auditory learning. And if true, they should certainly be published in a high-impact journal like Nature Human Behavior.

Below I make some suggestions for improvement in the manuscript, as well as some reservations about both the methods and interpretation.

1. The overall design of this study is based on Cheour et al. (2002) who showed that basic vowel discrimination can be obtained using EEG with newborns. And, in fact, there are many such demonstrations of vowel discrimination in early infancy. So the Introduction should not dwell on some failures of vowel discrimination which can be chalked up to poor methods or very subtle vowel distinctions (i.e., with low-salience acoustic cues).

Thank you for this suggestion. Rather than dwelling on a discussion of whether or not neonates can discriminate vowels at birth, the revised introduction now focuses on the fact that previous demonstrations of vowel discrimination at birth have not yet provided much information regarding the neural substrates involved and dynamic aspects of postnatal learning which is the purpose of the present study [see page 4, lines 65-67; also see page 12, lines 222-225].

2. The specific choice of vowel discrimination in the present study is quite unusual. The authors say they took great pains to use steady-state vowels (i.e., not vowels in consonant contexts) and to use only one token of each of 12 vowels. The vowel contrast, then, was not about a robust vowel category difference in Chinese, but rather a VERY subtle contrast between a forward vowel and its waveform reversal. It is not clear why the authors chose this contrast because steady-state vowels, by definition, have the SAME spectral information whether forward or reversed (and no prosodic differences as verified by adult ratings). However, upon closer examination (see Supplemental Fig. 4), there were two acoustic cues that differentiated the forward vs. reversed vowels – (1) a rapid vowel-onset and amplitude increase and a gradual vowel-offset and amplitude decrease in the forward tokens vs. the opposite pattern in the reverse tokens, and (2) the presence of vowel-onset glottalization in the forward tokens and vowel-offset glottalization in the reverse tokens. These two differences in the reverse tokens are not “natural” (as reflected in the ratings by adults), but that is not particularly relevant in the present design because prenatal acoustic exposure is unlikely to have provided the fetuses with these cues due to the low-pass filtering of the uterus. The main point here is that these are VERY subtle differences and therefore would be expected to be difficult to discriminate by newborns.

We agree that the stimuli used are particularly difficult to discriminate for newborns, and also share the reviewer’s enthusiasm for the fact that despite the subtlety of the contrast, we detected clear signs of learning within the first day of life. Our goal in this study was to explore how newborns learn to perceive steady-state vowels that are unlikely to have been (fully) acquired *in utero*. We chose to use single vowels because vowel discrimination in syllables (e.g., short CV) might be affected by voicing, manner, and place of articulation. One would have to pair the vowels with every possible consonant to show that discrimination of vowel quality is independent of consonant context in neonates. This would result in a much more complicated and lengthy experiment, possibly incompatible with newborn testing. Also, as

one of our own studies showed previously, neonates can dissociate sounds with different affective prosodies at birth (Zhang, Chen, Hou & Wu, 2019), and thus the choice of contrasting a forward vowel with its waveform reversal helps minimize prosodic variation as a potential acoustic cue. We now explain these points more clearly in the revised manuscript [see page 4, lines 77-79; also see page 14, lines 280-295].

3. The fact that only the experimental condition showed a learning effect after 5 hours of exposure implies that the active control group, who also listened to 5 hours of DIFFERENT vowels, were unable to extract the differences in amplitude onset/offset and glottalization onset/offset and generalize them to the test tokens. This is a remarkably NARROW example of learning (i.e., specific only to the vowels presented during the exposure phase).

We agree that results in the active control group show an example of narrow learning, suggesting that neonates, on their first day of life, can acquire and distinguish specific vowel sound which differ in a minimal fashion. This point is now emphasized in the revised manuscript [see page 12, lines 231-235].

4. The consolidation effect in the experimental group is quite interesting, especially since many phonetic learning effects in infants fail to be retained even after a few minutes delay.

We agree with the reviewer that the consolidation effect observed here is important and novel. As shown by the polysomnography data, neonates were mostly asleep in the gap between T1 and T2. Sleep might be a factor that previous studies have not systematically controlled for or reported. In the revised manuscript, we discussed links between sleep and memory formation as a possible account for the consolidation effect observed here [see page 14-15, lines 296-310].

In summary, you should emphasize the preceding four points because they set the stage for why these findings are so impressive.

We thank the reviewer for calling the points above to our attention. We have highlighted these four points in the revised manuscript, as we agree that they are important to spell out the impact of our findings.

5. fNIRS is not a terribly sensitive method for detecting subtle auditory discrimination abilities, especially in newborns. And the use of a latency measure is particularly unusual (Dehaene-Lambertz et al. 2006 used it with fMRI but that was with sentences, not isolated vowel syllables). For example, the really short latencies at T1 and T2 in the experimental group are highly suspect. I have never seen such rapid rises in the HRF in ANY fNIRS study, suggesting they are due to spontaneous background fluctuations in the underlying hemodynamics. I suggest moving these latency results to the Supplemental Materials.

Following the suggestion to use LMERs rather ANOVA (see below), we are in a position to confirm the claimed latency shifts via the new approach. Although researchers seldom consider the timecourse of fNIRS measurements, such analyses have long been reported to convey meaningful information in the case of other oxygenation-based measures with slow timecourses (e.g., time-resolved fMRI, see for instance Thierry et al., 1999). Our 10 Hz measurement rate provides a suitable basis for analyzing the peak latency of [HbO] over time (see revised Methods). Given the saliency of the latency effects and their importance in making a case of inferior frontal region engagement in vowel discrimination learning, we stand by our decision to include these results in the main text of the manuscript. We hope that the reviewer will find our new analysis convincing in this regard **[see page 8-9, lines 146-180 and New Figure 2]**.

6. You should conduct a linear mixed model to analyze your fNIRS amplitude effects rather than a set of ANOVAs.

Thank you for this suggestion. We have completely reanalysed our amplitude data, using linear mixed effects modelling to include channels as random effects in a single large model rather than using ANOVA to analyse each channel independently. We have also applied this same approach for our latency and connectivity analyses, and completely rewritten our results' section accordingly. While we believe the results of this new approach remain consistent with our original conclusions, we agree that a LMER analysis provides a much stronger empirical basis for theoretical inferencing **[see page 5-8, lines 113-145]**.

7. You should conduct a permutation test with your functional connectivity analyses to guard against false positives. How do we know that similar connectivity results were not present in other (i.e., non-language related) ROIs?

Thank you for raising this concern. In our new connectivity analysis, we have greatly de-emphasized the selection of specific channels, instead adapting the same LMER approach that we used for the amplitude and latency analyses. We first calculated correlations between every pair of channels for each subject at each time point, and then included all correlations in a single LMER model, including channel pair as a random effect. The model identifies a general cross-session increase in connectivity for the experimental group and active controls, compared to the passive controls **[see page 10-12, lines 181-220]**, and the **new Figure 3** illustrates the distribution of this increase.

8. You need to explain in more detail how you did anatomical co-registration of the fNIRS optodes. This is crucial for analyses that involve SINGLE fNIRS channels because they must sample the same underlying cortical regions across the 25 newborns in each of the three experimental conditions.

When the cap is set on the neonate participants' head, we measure the distance between the nose concave point and back head bulge point (Inion) to locate channel Cz, which is then marked with a white

dot on the cap. Cz, Nz (nose concave point), Iz (Inion), AR (above the right ear), and AL (above the left ear) are used as positional references to ensure that the cap is correctly set on the participant's head. This procedure reduces variations in single fNIRS channels placement and increases the chances that the same underlying cortical regions are targeted across participants. After cap placement, fNIRS channels location is registered using a 3D digitizer (Patriot, Polhemus, VT, USA) in reference to a neonatal head model (Shi et al., 2011), in each participant. This procedure is now described in detail in the revised manuscript [see page 22-23, lines 557-570].

9. I found your General Discussion about possible motor involvement in these findings to be highly speculative. You would need a different experimental design to confirm that interpretation.

We agree with the Reviewer that our claim regarding motor involvement was somewhat speculative and that it would require a dedicated experimental design to test. In the revised manuscript, we have substantially toned down this proposed interpretation and we have removed the last paragraph of the Discussion where the idea was mostly developed (i.e., subvocal imitation). We now propose that our study provides evidence for the neural connectivity between sensory, motor, and somatosensory brain areas, which might support integration of sensory and motor speech processing from the very beginning of an infant's life. Future studies might investigate how the neural network characterized in the present study could be the foundation of a sensorimotor network involved in speech perception in later life [see page 16, lines 352-368; also see page 17, lines 376-381].

Decision Letter, first revision:

17th February 2022

Dear Professor Zhang,

Thank you once again for your revised manuscript, entitled "Sensory-motor neural network supports ultra-fast tuning to natural vowels in neonates", and for your patience during the peer review process.

Your Article has now been reviewed again by 2 referees. You will see from their comments copied below that, although they find your work improved, Reviewer #2 continues to express substantial concerns. In light of these comments, we cannot accept the manuscript for publication, but would be interested in considering another revised version if you are willing and able to fully address reviewer and editorial concerns.

We hope you will find the referees' comments useful as you decide how to proceed. If you wish to submit a substantially revised manuscript, please bear in mind that we would be unable to proceed further with the work unless Reviewer #2's concerns are fully addressed and the reviewer is satisfied with the revision. We view this as the final chance to address Reviewer #2's concerns over the fNIRS analysis.

If you wish to submit a suitably revised manuscript we would hope to receive it within 6 months. We understand that the COVID-19 pandemic is causing significant disruptions which may prevent you from carrying out the additional work required for resubmission of your manuscript within this timeframe. If you are unable to submit your revised manuscript within 6 months, please let us know. We will be happy to extend the submission date to enable you to complete your work on the revision.

- Include a "Response to the editors and reviewers" document detailing, point-by-point, how you addressed each editor and referee comment. If no action was taken to address a point, you must provide a compelling argument. This response will be used by the editors to evaluate your revision and sent back to the reviewers along with the revised manuscript.
- Highlight all changes made to your manuscript or provide us with a version that tracks changes.

[REDACTED]

Thank you for the opportunity to review your work. Please do not hesitate to contact me if you have any questions or would like to discuss the required revisions further.

Sincerely,
Jamie

Dr Jamie Horder
Senior Editor
Nature Human Behaviour

REVIEWER COMMENTS:

Reviewer #1:

Remarks to the Author:

I believe that the authors have done an excellent job in responding to the reviewers, making careful and thoughtful edits to focus the manuscript on the most novel and important aspects of their findings. It's an important contribution to the literature and I have recommended it for priority publication in Nature Human Behavior.

There is just one place where I would ask the authors to correct the text. On line 354, the authors quote Kuhl (81), saying that she put forward the proposal that infants are equipped with "an internal, innately specified vocal-tract synthesizer." In Kuhl's chapter, she uses that quote (see page 144), but she is citing that quote from Liberman and Mattingly's 1985 paper in Cognition (vol 21, pages 1-36). It is Liberman and Mattingly's interpretation that Analysis by Synthesis (AxS) stipulates that infants are born with an internal innately specified vocal-tract synthesizer.

What Kuhl states as her own position on Analysis by Synthesis is different, and I quote here from Kuhl in that same paragraph, "The alternative view takes a different position on AxS, suggesting that neural connectivity between auditory and motor areas exists at birth, allowing information to flow between them....This is conceived as a learning model, driven by experience, in which the innate component is neural connectivity that links information from sensory, motor, and somatosensory brain areas." (page 144).

Thus Kuhl's position is that connectivity exists at birth, and that early experience of speech allows immediate sensorimotor learning. The present study is totally compatible with this idea. It's not essential to posit a vocal track synthesizer that exists at birth, but only to posit that innate connectivity allows rapid and efficient communication between sensory and motor brain areas, which in turn allows sensorimotor learning, based on experience.

Patricia K. Kuhl

Reviewer #2:

Remarks to the Author:

The authors have done a very nice job of responding to the original round of reviews. The manuscript is now more focused on the effects of vowel learning, and the mixed effects models for amplitude and latency are quite nice.

But I am not convinced that the functional connectivity (FC) analyses are conducted properly given the constraints of the fNIRS recordings. The primary problem with FC metrics with fNIRS recordings is that there are many "false" correlations due to surface vascular noise (i.e., non-cortical signals). There are two ways to reduce these spurious surface correlations: short channel regression and spatial filtering. You did not have short channels and spatial filtering COULD be used but it is not clear in the literature with newborns whether it is a suitable way to "clean up" the 2.3 cm channels that are your main dependent measure.

My suggestion, therefore, is to take a more conservative and focused approach in analyzing the FC data. Rather than looking at all pairwise channel correlations (most of which are not significant), focus on the statistically significant channels from your amplitude and latency results. Use these channels as "seeds" and ask how the network of pairwise correlations with all the other channels differs between the three groups of subjects.

In addition, only use channels as seeds that are significant AFTER correcting for multiple comparisons (i.e., the UNCORRECTED $p < .05$ is not stringent enough). This might be only channels 7 and 45 for amplitude and channels 2,6, 43, and 44 for latency.

Author Rebuttal, first revision:

Point-by-point reply to Reviewers' comments

Reviewer #1:

Remarks to the Author:

I believe that the authors have done an excellent job in responding to the reviewers, making careful and thoughtful edits to focus the manuscript on the most novel and important aspects of their findings. It's an important contribution to the literature and I have recommended it for priority publication in Nature Human Behavior.

There is just one place where I would ask the authors to correct the text. On line 354, the authors quote Kuhl (81), saying that she put forward the proposal that infants are equipped with "an internal, innately specified vocal-tract synthesizer." In Kuhl's chapter, she uses that quote (see page 144), but she is citing that quote from Liberman and Mattingly's 1985 paper in Cognition (vol 21, pages 1-36). It is Liberman and Mattingly's interpretation that Analysis by Synthesis (AxS) stipulates that infants are born with an internal innately specified vocal-tract synthesizer.

What Kuhl states as her own position on Analysis by Synthesis is different, and I quote here from Kuhl in that same paragraph, "The alternative view takes a different position on AxS, suggesting that neural connectivity between auditory and motor areas exists at birth, allowing information to flow between them.... This is conceived as a learning model, driven by experience, in which the innate component is neural connectivity that links information from sensory, motor, and somatosensory brain areas." (page 144).

Thus Kuhl's position is that connectivity exists at birth, and that early experience of speech allows immediate sensorimotor learning. The present study is totally compatible with this idea. It's not essential to posit a vocal track synthesizer that exists at birth, but only to posit that innate connectivity allows rapid and efficient communication between sensory and motor brain areas, which in turn allows sensorimotor learning, based on experience.

We acknowledge the misrepresentation of the Reviewer's position regarding the AxS hypothesis and we are grateful for the correction. This passage now reads as follows: "Recently, Kuhl⁸¹ proposed that innate neural connectivity between sensory, motor, and somatosensory brain areas equips infants with an experience-driven sensorimotor learning device allowing learning to begin immediately after birth (cf. the analysis by synthesis model).^{82,83}" [see page 16, lines 368-371].

Reviewer #2:

Remarks to the Author:

The authors have done a very nice job of responding to the original round of reviews. The manuscript is now more focused on the effects of vowel learning, and the mixed effects models for amplitude and latency are quite nice.

But I am not convinced that the functional connectivity (FC) analyses are conducted properly given the constraints of the fNIRS recordings. The primary problem with FC metrics with fNIRS recordings is that there are many "false" correlations due to surface vascular noise (i.e., non-cortical signals). There are two ways to reduce these spurious surface correlations: short channel regression and spatial filtering. You did not have short channels and spatial filtering COULD be used but it is not clear in the literature with newborns whether it is a suitable way to "clean up" the 2.3 cm channels that are your main dependent measure.

My suggestion, therefore, is to take a more conservative and focused approach in analyzing the FC data. Rather than looking at all pairwise channel correlations (most of which are not significant), focus on the statistically significant channels from your amplitude and latency results. Use these channels as "seeds" and ask how the network of pairwise correlations with all the other channels differs between the three groups of subjects.

In addition, only use channels as seeds that are significant AFTER correcting for multiple comparisons (i.e., the UNCORRECTED $p < .05$ is not stringent enough). This might be only channels 7 and 45 for amplitude and channels 2, 6, 43, and 44 for latency.

We acknowledge that surface vascular noise could potentially affect the functional connectivity (FC) analyses, and we agree that it may therefore be useful to focus our connectivity analyses on a subset of known-relevant channels. In our revision, we have therefore restricted our FC analyses to 7 'seed' channels, {2, 6, 7, 10, 43, 44, 45}, which we identified by applying a False Discovery Rate procedure to the per-channel p-values from the amplitude and latency analyses. Note that we apply a slightly higher FDR threshold than normal ($q < .15$; e.g., Benjamini & Hochberg, 1995; Genovese et al., 2002), as a

compromise because (1) our predictions are directional, and (2) standard FDR corrections for a set of analyses are excessively conservative when one can already reject the null hypothesis for the group, implying $p_0 < 1$. The results of the new FC analyses are very similar to those of our previous analyses. For the cross-channel model, the crucial Group \times Phase interaction in the new analysis is estimated as $b=0.217$, $p=.001$; in the previous analysis it was estimated as $b=0.161$, $p=.001$; if applying a stronger FDR correction ($q < .05$, thus restricting the seed channels to {2, 6, 43, 44}), the crucial interaction still remains essentially the same: $b=0.221$, $p=.002$.

Estimates for the individual correlations are, of course, unchanged. In the revised manuscript, we have updated the Results with the restricted FC analyses [see page 10-12; also see page 22 for updated methods]. We have also clarified that we consider the FC analyses to be an exploratory extension of our study, and that the main conclusions are based on the amplitude and latency results [see page 15-16, lines 340-358].

Decision Letter, second revision:

Our ref: NATHUMBEHAV-210716090B

23rd March 2022

Dear Dr. Zhang,

Thank you for submitting your revised manuscript "Sensory-motor neural network supports ultra-fast tuning to natural vowels in neonates" (NATHUMBEHAV-210716090B). It has now been seen by the original referees and their comments are below. As you can see, the reviewers find that the paper has improved in revision.

We will therefore be happy in principle to publish it in Nature Human Behaviour, pending minor revisions to comply with our editorial and formatting guidelines.

We are now performing detailed checks on your paper and will send you a checklist detailing our editorial and formatting requirements within a week. Please do not upload the final materials and make any revisions until you receive this additional information from us.

Sincerely,
Jamie

Dr Jamie Horder
Senior Editor
Nature Human Behaviour

Reviewer #2 (Remarks to the Author):

The authors have done an excellent job of responding to the two reviewers' comments. I asked for a revised functional connectivity analysis and was pleased to see that my suggestion was followed, and more importantly, that it was found to support the authors' original findings.

The only minor suggestion I would make before final acceptance is to move Table 1 to the Supplementary materials. It is just too many correlations for the main text.

Author Rebuttal, second revision:

Point-by-point reply to Reviewers' comments

Reviewer #2:

Remarks to the Author:

The authors have done an excellent job of responding to the two reviewers' comments. I asked for a revised functional connectivity analysis and was pleased to see that my suggestion was followed, and more importantly, that it was found to support the authors' original findings.

The only minor suggestion I would make before final acceptance is to move Table 1 to the Supplementary materials. It is just too many correlations for the main text.

The Table 1 is now moved to Supplementary materials (Table S1).

Final Decision Letter:

Dear Professor Zhang,

We are pleased to inform you that your Article "Rapid learning of a phonemic discrimination in the first hours of life", has now been accepted for publication in Nature Human Behaviour.

Please note that *Nature Human Behaviour* is a Transformative Journal (TJ). Authors whose manuscript was submitted on or after January 1st, 2021, may publish their research with us through the traditional subscription access route or make their paper immediately open access through payment of an article-processing charge (APC). Authors will not be required to make a final decision about access to their article until it has been accepted. IMPORTANT NOTE: Articles submitted before January 1st, 2021, are not eligible for Open Access publication. Find out more about Transformative Journals

We welcome the submission of potential cover material (including a short caption of around 40 words) related to your manuscript; suggestions should be sent to Nature Human Behaviour as

electronic files (the image should be 300 dpi at 210 x 297 mm in either TIFF or JPEG format). Please note that such pictures should be selected more for their aesthetic appeal than for their scientific content, and that colour images work better than black and white or grayscale images. Please do not try to design a cover with the Nature Human Behaviour logo etc., and please do not submit composites of images related to your work. I am sure you will understand that we cannot make any promise as to whether any of your suggestions might be selected for the cover of the journal.

With best regards,

Jamie

Dr Jamie Horder

Senior Editor
Nature Human Behaviour